# Agonist-specific voltage-dependent gating of lysosomal two-pore Na+ channels

Xiaoli Zhang[1†]*, Wei Chen[1†], Ping Li[1,2†], Raul Calvo[3], Noel Southall[3], Xin Hu[3], Melanie Bryant-Genevier[3], Xinghua Feng[2], Qi Geng[1], Chenlang Gao[1], Meimei Yang[1,4], Kaiyuan Tang[1], Marc Ferrer[3], Juan Jose Marugan[3], Haoxing Xu[1]*

[1]Department of Molecular, Cellular, and Developmental Biology, University of Michigan, Ann Arbor, United States; [2]Collaborative Innovation Center of Yangtze River Delta Region Green Pharmaceuticals, Zhejiang University of Technology, Hangzhou, China; [3]National Center for Advancing Translational Sciences (NCATS), Medical Center Drive, Rockville, United States; [4]Department of Neurology, The Fourth Hospital of Harbin Medical University, Harbin, China

**Abstract** Mammalian two-pore-channels (TPC1, 2; *TPCN1, TPCN2*) are ubiquitously- expressed, $PI(3,5)P_2$-activated, $Na^+$-selective channels in the endosomes and lysosomes that regulate luminal pH homeostasis, membrane trafficking, and *Ebola* viral infection. Whereas the channel activity of TPC1 is strongly dependent on membrane voltage, TPC2 lacks such voltage dependence despite the presence of the presumed 'S4 voltage-sensing' domains. By performing high-throughput screening followed by lysosomal electrophysiology, here we identified a class of tricyclic anti-depressants (TCAs) as small-molecule agonists of TPC channels. TCAs activate both TPC1 and TPC2 in a voltage-dependent manner, referred to as Lysosomal Na+ channel Voltage-dependent Activators (LyNa-VAs). We also identified another compound which, like $PI(3,5)P_2$, activates TPC2 independent of voltage, suggesting the existence of agonist-specific gating mechanisms. Our identification of small-molecule TPC agonists should facilitate the studies of the cell biological roles of TPCs and can also readily explain the reported effects of TCAs in the modulation of autophagy and lysosomal functions.

**\*For correspondence:**
zhangxi@umich.edu (XZ);
haoxingx@umich.edu (HX)

[†]These authors contributed equally to this work

**Competing interests:** The authors declare that no competing interests exist.

## Introduction

Lysosomes are the cell's recycling center equipped with the most important nutrient-sensing machinery in the cell (*Lawrence and Zoncu, 2019*; *Li et al., 2019*). Ion channels in the lysosome play essential roles in the regulation of various lysosomal functions, including cargo import, lysosomal degradation, and catabolite export (*Li et al., 2019*; *Xiong and Zhu, 2016*). Patch-clamp studies of isolated lysosomal membranes have recently discovered multiple lysosomal channels that are selective for $Na^+$, $K^+$, $Ca^{2+}$, and $Cl^-$ (*Cang et al., 2015*; *Cang et al., 2013*; *Cao et al., 2015*; *Dong et al., 2010*; *Li et al., 2019*; *Wang et al., 2012*; *Xiong and Zhu, 2016*). How these lysosomal channels are activated by endogenous nutrient-dependent signals remain largely unknown (*Li et al., 2019*). On the other hand, membrane-permeable small-molecule modulators, that is synthetic agonists and inhibitors, have proved extremely helpful in probing the cell biological functions of intracellular channels, including lysosomal channels (*Cao et al., 2015*; *Dong et al., 2010*; *Li et al., 2019*; *Wang et al., 2012*; *Xiong and Zhu, 2016*). For example, small-molecule synthetic agonists of Mucolipin TRP channels (TRPMLs), that is ML-SAs, have been instrumental in revealing the functions of these $Ca^{2+}$ release channels in lysosomal exocytosis, mobility, and biogenesis (*Li et al., 2016*; *Shen et al.,*

*2012*; *Zhang et al., 2016*). However, such chemical tools are still lacking for most other lysosomal channels.

Two-pore channel proteins (TPC1, 2; *TPCN1, TPCN2*) are ubiquitously expressed, dimeric, two-repeat ($2 \times 6$ TM) cation channels that are localized exclusively in the intracellular endosomes and lysosomes (*Calcraft et al., 2009*; *Grimm et al., 2017*; *Morgan et al., 2011*). At the cellular level, TPCs regulate organellar membrane excitability, membrane trafficking, and pH homeostasis; at the organismal level, TPCs regulate various physiological and pathological processes, including hair pigmentation, *Ebola* viral infection, and cancer growth (*Ambrosio et al., 2016*; *Nguyen et al., 2017*; *Sakurai et al., 2015*). Early works from several laboratories suggested that TPCs play an essential role in mediating $Ca^{2+}$ release from endolysosomes in response to cytosolic increases of nicotinic acid adenine dinucleotide phosphate (NAADP) (*Brailoiu et al., 2009*; *Calcraft et al., 2009*; *Ruas et al., 2010*; *Zong et al., 2009*). However, it remains controversial whether TPCs are the *bona fide* NAADP receptor (*Lin-Moshier et al., 2012*; *Morgan et al., 2015*; *Walseth et al., 2012*). Indeed, several recent endolysosomal patch-clamp studies have demonstrated that TPCs are $Na^+$-selective channels activated by PI(3,5)P$_2$ (*Bellono et al., 2016*; *Cang et al., 2014*; *Cang et al., 2013*; *Guo et al., 2017*; *Jha et al., 2014*; *Kirsch et al., 2018*; *She et al., 2019*; *Wang et al., 2012*), a late endosome and lysosome-specific phosphoinositide that is known to regulate many aspects of lysosome function (*McCartney et al., 2014*). Recent high-resolution structural studies revealed that several amino acid (AA) residues in the selectivity filter of TPCs confer the selectivity of $Na^+$ over $K^+$ or $Ca^{2+}$ (*Guo et al., 2016*; *Guo et al., 2017*), and that PI(3,5)P$_2$ binds directly to several positively charged AA residues in the S4-S5 linker to induce channel opening (*Kirsch et al., 2018*; *She et al., 2018*; *She et al., 2019*). Sphingosines also reportedly induce TPC1-mediated $Ca^{2+}$ release from the lysosomes (*Höglinger et al., 2015*), but direct activation of TPCs by sphingosines was not confirmed in the lysosomal electrophysiological assays (unpublished data in the Xu laboratory) (*Li et al., 2019*).

Lysosomal membrane potential ($\Delta\psi$) has been proposed to regulate an array of lysosomal functions, including metabolite transport and membrane trafficking, but the underlying mechanisms are poorly understood (*Li et al., 2019*; *Xu and Ren, 2015*). $Na^+$ flux mediated by TPCs may cause rapid changes of lysosomal $\Delta\psi$, which may in turn modulate the functions of TPCs and other lysosomal channels (*Cang et al., 2014*). Like canonical voltage-gated cation channels, TPCs contain multiple positively-charged AA residues in their voltage sensor domains (S4), which are believed to confer voltage-dependent activation of plant and animal TPC1 channels (*Cang et al., 2014*; *She et al., 2019*). In a sharp contrast, TPC2 activation is completely voltage-independent, despite the presence of multiple Arginine/Lysine residues in their S4 helices (*Cang et al., 2014*; *Cang et al., 2013*; *Wang et al., 2012*). In the current study, we identified seven small molecules known to act as tricyclic anti-depressants (TCAs) that activate both TPC1 and TPC2 in a voltage-dependent manner.

## Results

### High-throughput screening of small-molecule agonists of TPC2 channels

We recently used a $Ca^{2+}$-imaging-based high-throughput screening (HTS) method to identify small-molecule agonists for lysosomal TRPML1 channels (*Wang et al., 2015*). Although TPCs are $Na^+$-selective channels with limited $Ca^{2+}$ permeability (*Cang et al., 2014*; *Guo et al., 2017*; *Li et al., 2019*; *Wang et al., 2012*), in a number of cell-based studies, TPCs reportedly mediate $Ca^{2+}$ release from lysosomes (*Brailoiu et al., 2009*; *Calcraft et al., 2009*; *Jha et al., 2014*; *Morgan et al., 2015*; *Patel, 2015*; *Ruas et al., 2015*; *Zong et al., 2009*), and it is conceivable that the small $Ca^{2+}$-permeability of TPCs, or activation of $Na^+$-dependent $Ca^{2+}$ flux mechanisms (e.g., $Na^+$-$Ca^{2+}$ exchanger) secondary to $Na^+$ flux may be sufficient to elevate intracellular $Ca^{2+}$. We thus screened HEK293 cells stably expressing human TPC2 (hTPC2) channels with the Library of Pharmacologically Active Compounds (LOPAC) (*Liu et al., 2010*), the same library of chemicals that were previously tested on TRPML1 channels. Among the positive hits, 23 compounds induced $Ca^{2+}$ increases in TPC2 stable cells (*Figure 1A and B* and *Figure 1—figure supplement 1A*), but not in cells stably expressing TRPML1[4A] (a surface-expressing mutant TRPML1 [*Shen et al., 2012*]) channels (data not shown).

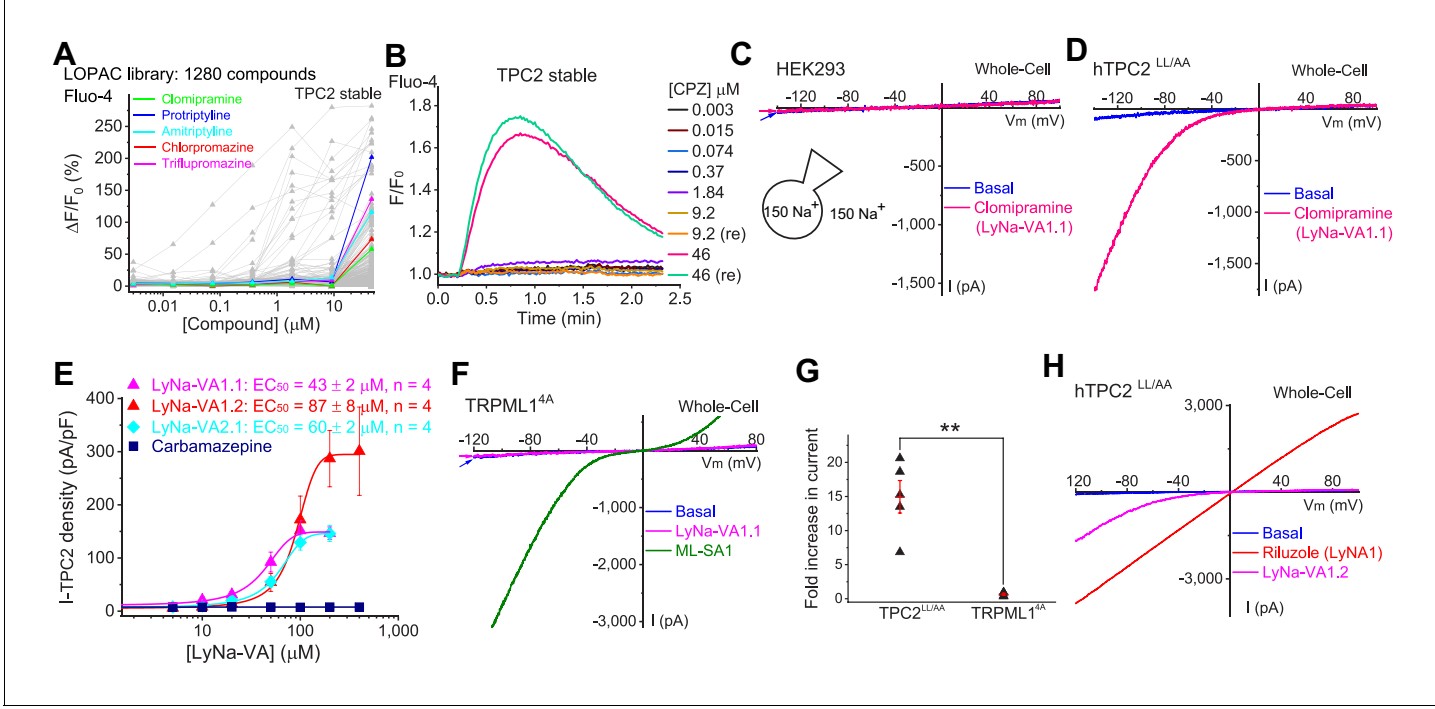

**Figure 1.** Screening of small-molecule agonists of TPC2. (A) High-throughput screening of the LOPAC library with Fluo-4 $Ca^{2+}$ imaging in HEK293 cells stably expressing hTPC2 (Dryad, http://doi.org/10.5061/dryad.s5f6j9h). Each trace represented the average $Ca^{2+}$ response to individual LOPAC compound. Only positive hits confirmed with electrophysiology were color-coded. (B) An example of a positive responder (chlorpromazine, CPZ), which elevated intracellular $Ca^{2+}$ levels at the concentration of 46 μM. Note a similar response was seen with a repeated (re) drug application. (C) Representative whole-cell currents in a HEK293 cell upon bath application of clomipramine (100 μM). Both pipette and bath solutions contained symmetric 150 mM $Na^+$. Currents were elicited by repeated voltage ramps (−140 to +100 mV; 200 ms) with a 4 s inter-step interval. Holding potential (HP) = 0 mV. (D) Representative TPC2-mediated currents ($I_{TPC2}$) activated by clomipramine (100 μM; LyNa-VA1.1) in a HEK293 cell transfected with a surface-expressing mutant TPC2 channel (EGFP-TPC2$^{LL/AA}$; *Wang et al., 2012*). (E) Dose-dependent activation of TPC2 by Lysosomal $Na^+$ channel Voltage-dependent Activators (LyNa-VAs). (F) The effect of clomipramine (100 μM) on surface-expressing mutant TRPML1 channels (TRPML1$^{4A}$; *Shen et al., 2012*). (G) Summary of clomipramine effects on whole-cell $I_{TPC2-LL/AA}$ and $I_{TRPML1-4A}$. Individual data and Mean ± S.E.M are presented. **, p<0.01. (H) Activation of $I_{TPC2}$ by LyNa-VA1.2 and Lysosomal $Na^+$ channel Agonist 1 (LyNA1; see *Figure 1—figure supplement 2*). Individual data for (A), (E) and (G) are presented in *Figure 1—source data 1*.

The online version of this article includes the following source data and figure supplement(s) for figure 1:

**Source data 1.** Screening of small-molecule agonists of TPC2.
**Figure supplement 1.** TCAs act as TPC agonists.
**Figure supplement 2.** Time courses of LyNa-VA1.2-induced TPC2 activation under different recording configurations.
**Figure supplement 2—source data 1.** Time course of LyNa-VA1.2- induced $I_{TPC2}$.
**Figure supplement 3.** LyNA1 activates TPC2 channels.

## Tri-cyclic anti-depressants (TCAs) as TPC2 agonists

To our surprise, 5 out of the 23 compounds are well characterized as tri-cyclic anti-depressants (TCAs), which are believed to act on neurotransmitter transporters or voltage-gated $Na^+$ channels (*Cheng and Bahar, 2019*; *Pancrazio et al., 1998*). We therefore characterized the responses of TPC2 channels to TCAs in detail using electrophysiological methods. We first performed whole-cell recordings in HEK293 cells that were transfected with surface-expressed TPC2 mutant channels (TPC2-L$^{11}$L$^{12}$-AA; abbreviated as TPC2$^{LL/AA}$ hereafter). No functional difference was noted between wild-type (WT) TPC2 and TPC2$^{LL/AA}$ channels in terms of channel permeation and gating properties (*Wang et al., 2012*). To facilitate the detection of TPC2-specific currents, we used symmetric $Na^+$ solutions in the bath (extracellular) and pipette (cytosolic) solutions (*Wang et al., 2012*). All five TCAs robustly and rapidly activated whole-cell currents in TPC2$^{LL/AA}$-expressing cells (*Wang et al., 2012*), but not in non-transfected HEK293 cells (*Figure 1C and D*, *Figure 1—figure supplement 1B–D*, *Table 1*, and *Table 1—source data 1*).

**Table 1.** Summary of electrophysiology-based screening of TPC agonists.

Based on chemical structures, LyNa-VAs were divided into two groups: LyNa-VA1.x and LyNa-VA2.x. $EC_{50}$ and the average TPC2 currents ($I_{TPC2}$) were calculated based on 3–5 whole-cell or whole-endolysosome recordings (see individual data in *Table 1—source data 1*) for each LyNa-VA, respectively.

| LyNa-VAs | Chemical name | Structure | $EC_{50}$ (µM)* | $I_{TPC2}$ (pA)** |
|---|---|---|---|---|
| LyNa-VA1.1 | Clomipramine |  | 43 ± 2 | 945 ± 111 |
| LyNa-VA1.2 | Desipramine |  | 87 ± 8 | 1120 ± 94 |
| LyNa-VA1.3 | Imipramine |  | 112 ± 1 | 433 ± 94 |
| LyNa-VA1.4 | Amitriptyline |  | 102 ± 3 | 876 ± 196 |
| LyNa-VA1.5 | Nortriptyline |  | 52 ± 10 | 1916 ± 361 |
| | Carbamazepine |  | No activation | No activation |
| LyNa-VA2.1 | Chlorpromazine |  | 60 ± 2 | 1101 ± 508 |
| LyNa-VA2.2 | Triflupromazine |  | 63 ± 2 | 984 ± 294 |
| | Phenothiazine |  | No activation | No activation |

*Data were obtained from whole-cell recordings at −140 mV.

**Data were obtained from whole endolysosome recordings with 100 µM of LyNa-VAs at −120 mV.

The online version of this article includes the following source data for Table 1:

**Source data 1.** Electrophysiology-based screening of TPC agonists.

We then extended detailed analyses to other known TCAs (*Table 1*). Of them, Clomipramine activated whole-cell TPC2$^{LL/AA}$-mediated strongly rectifying currents ($I_{TPC2-LL/AA}$) with an EC$_{50}$ of $43 \pm 2$ µM (n = 4 patches) (*Figure 1D and E*, and *Table 1*). Given the apparent voltage-dependent gating described below, we referred to Clomipramine as Lysosomal Na$^+$ channel Voltage-dependent Activator 1.1 (LyNa-VA1.1). Another structurally different TCA, Chlorpromazine, had an EC$_{50}$ of $60 \pm 2$ µM (n = 4 patches), and was referred to as LyNa-VA2.1 (*Figure 1E*, *Table 1*, and *Figure 1—figure supplement 1B*). Other TPC-activating TCAs were referred to as LyNa-VA1.x or LyNa-VA2.x, respectively, based on the structural similarity (*Table 1*). In contrast, no significant activation was seen with Carbamazepine or Phenothiazine, tricyclic drugs without the aliphatic chain (*Table 1* and *Figure 1—figure supplement 1D*). All TCA-induced currents exhibited strong voltage-dependence and inward rectification, which resembled TRPML1-mediated currents ($I_{TRPML1}$) (*Shen et al., 2012*). However, none of the TCAs had any activation effect on $I_{TRPML1}$ (*Figure 1F and G*). Taken together, these results suggested that TCAs may function as small-molecule agonists of TPC2 channels.

In a separate screen, we identified another compound (Riluzole, see *Figure 1—figure supplement 3A*) that showed striking differences from the responses elicited by TCAs. Riluzole, an FDA-approved amyotrophic lateral sclerosis drug that is known to modulate voltage-gated Na$^+$ channels (*Liu and Wang, 2018*), activated large and linear whole-cell currents in TPC2$^{LL/AA}$-expressing HEK293 cells (*Figure 1H* and *Figure 1—figure supplement 3*), but not in non-transfected HEK293 cells. Given its lack of the voltage-dependence, Riluzole was referred to as Lysosomal Na$^+$ channel Agonist 1 (LyNA1). Hence, more than one agonist-specific gating (voltage-dependent and voltage-independent) mechanism may co-exist within one channel protein.

## TCAs activate lysosomal TPC2 channels

TCAs, with a general structure of an aromatic greasy core and an aliphatic chain containing a terminal amine, are lysosomotropic compounds that are known to be highly accumulating in the lysosomes due to proton trapping (*Beckmann et al., 2014*). We next tested the effects of LyNa-VAs in hTPC2-transfected Cos1 and HEK293 cells using whole-endolyosome patch-clamp. Cells were pre-treated with vacuolin-1 (1 µM) that can selectively increase the size of late endosomes and lysosomes (LELs) up to 5 µm (*Dong et al., 2010*). The enlarged endolysosomes were manually isolated and then patch clamped in the whole-endolysosome configuration (*Dong et al., 2010*). In TPC2-positive enlarged LELs isolated from transfected Cos1 cells, little or no basal whole-endolysosome currents were seen under symmetric (pipette/luminal and bath/cytosolic) Na$^+$ solutions (*Figure 2A*). Consistent with our previous studies, bath application of PI(3,5)P$_2$, the endogenous agonist of TPCs (*Wang et al., 2012*), activated a large whole-endolysosome $I_{TPC2}$ with linear I-V (*Figure 2B and E*). Both LyNa-VAs and LyNA1 readily activated $I_{TPC2}$, although the I-Vs were dramatically different (*Figure 2C, D and E*, and *Figure 2—figure supplements 1* and *2*). In contrast, no current activation was seen for LyNa-VAs or LyNA1 in TRPML1-transfected cells (*Figure 2—figure supplement 1C*). Both LyNa-VAs and LyNA1 activated whole-endolysosome $I_{TPC}$ in WT HAP1 cells, but not HAP1 cells lacking *TPC1* or *TPC2* (*TPC1$^{-/-}$/TPC2$^{-/-}$*, TPC1/2 DKO; *Figure 2—figure supplement 1*), suggesting that the activation effects of both TCAs and LyNA1 on TPCs are specific. In addition, robust activation of $I_{TPC2}$ in the whole-cell configuration with agonists being applied from the extracellular side (analogous to the luminal side in the lysosome), as well as in the inside-out (*Figure 1—figure supplement 2A*) and whole-endolysosome configurations with agonists being applied from the cytosolic side, suggests that LyNa-VAs and LyNA1 are likely to be membrane permeable, activating TPC2 via direct agonist binding. As the channel activation in the whole-cell recordings was significantly slower, that is longer latency and time course of activation, compared to the inside-out and whole-endolysosome recordings (see *Figure 1—figure supplement 2*), the action site of LyNa-VAs is likely to be either intracellular or more accessible from the intracellular side.

## Activation of TPCs by LyNa-VAs shows strong voltage dependence

Despite the fact that TPCs have two S4 domains that act as voltage sensors in many voltage gated channels (*She et al., 2018*), the responses of TPC2 to PI(3,5)P$_2$ and LyNA1 were voltage independent, with large currents at both positive and negative voltages (*Figure 2B, D and E*). In contrast, TPC2 currents activated by TCAs were strongly voltage-dependent, in both whole-cell and whole-

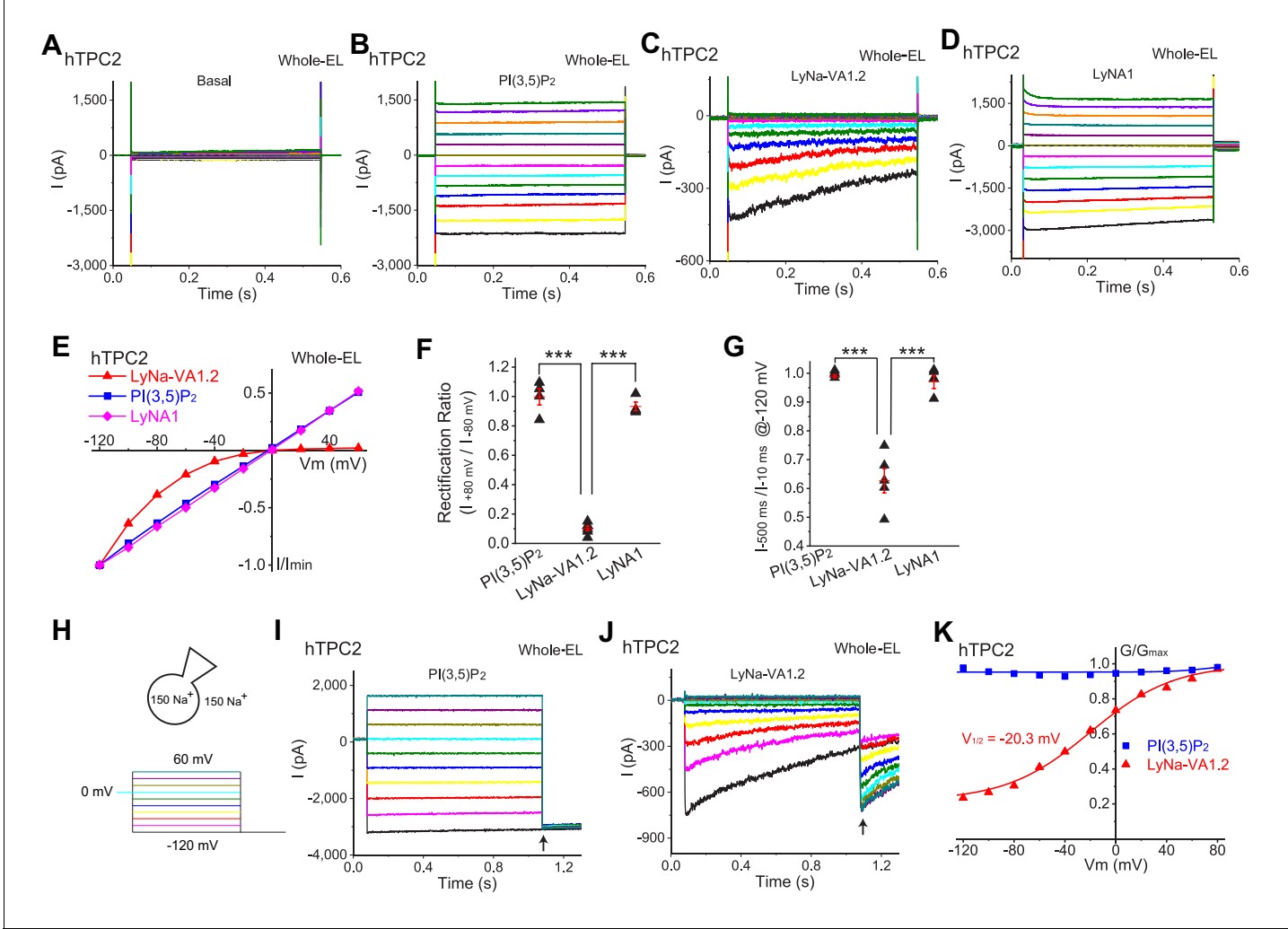

**Figure 2.** LyNa-VAs activate lysosomal TPC2 channels in a voltage-dependent manner. (A) Representative basal $I_{TPC2}$ step currents elicited by a voltage step protocol in the whole-endolysosome (EL) configuration. Voltage steps from −140 to 100 mV with a voltage increment (ΔV) of 20 mV for 0.5 s were used to elicit $I_{TPC2}$ in A-D. HP = 0 mV. Unless otherwise indicated, symmetric (bath/cytosol vs pipette/lumen) $Na^+$ (150 mM) solutions were used for all whole-endolysosome recordings, and PI(3,5)$P_2$ (0.3 μM), LyNa-VA1.2 (100 μM), and LyNA1 (300 μM) were bath- applied to induce $I_{TPC2}$. (B–D) Representative $I_{TPC2}$ step currents activated by PI(3,5)$P_2$ (B), LyNa-VA1.2 (C), and LyNA1 (D). (E) Representative normalized I-V plots based on the instantaneous currents activated by various agonists. (F) Rectification index, calculated as the ratio of the current amplitudes between +80 and −80 mV, of PI(3,5)$P_2$-, LyNa-VA1.2-, and LyNA1- activated $I_{TPC2}$. (G) The inactivation of $I_{TPC2}$ at −120 mV was quantified as the ratio of current amplitudes at 10 vs. 500 ms, based on step currents in B, (C) and D. (H) Voltage steps from −120 to 100 mV (ΔV = 20 mV) for 1 s were used to elicit tail currents at −120 mV shown in (I) and (J). (I) The tail currents of PI(3,5)$P_2$- evoked whole-endolysosome $I_{TPC2}$. (J) The tail currents of LyNa-VA1.2- activated whole-endolysosome $I_{TPC2}$. Arrows in (I) and (J) indicate where the currents were measured to calculate the channel conductance (G = I/V). (K) Normalized G-V curves of PI(3,5)$P_2$- and LyNa-VA1.2- activated $I_{TPC2}$. LyNa-VA1.2 activated $I_{TPC2}$ in a voltage dependent manner with a $V_{1/2}$ = −20.3 ± 3.5 mV (n = 5 patches). For panels F and G, individual data and Mean ± S.E.M. are presented. ***, p<0.001. Individual data for (F) and (G) are presented in *Figure 2—source data 1*.

The online version of this article includes the following source data and figure supplement(s) for figure 2:

**Source data 1.** The inactivation of $I_{TPC2}$ and rectification index of TPC2.
**Figure supplement 1.** LyNA1 activates lysosomal TPC2 channels.
**Figure supplement 2.** TCAs induce voltage-dependent activation and inactivation of TPC2 channels.
**Figure supplement 2—source data 1.** Agonist-specific voltage-dependent inactivation of $I_{TPC2}$ induced by LyNa-VA1.2 and LyNA1.

endolysosome recordings, no matter whether the currents were elicited by a voltage ramp or a series of voltage steps (see *Figure 2C and E* and *Figure 2—figure supplement 2*). In both whole-cell and whole-endolysosome configurations, TCA-activated step currents displayed prominent inactivation and strong inward rectification (*Figure 2C, E and F* and *Figure 2—figure supplement 2*), suggesting the existence of voltage-dependent activation and/or inactivation gating processes. The steady-state voltage dependence of the responses to all 7 TCAs were qualitatively similar. A convenient way to quantitatively summarize the voltage dependence of the TCA responses was to calculate the rectification ratio ($\frac{I@+80mV}{I@-80mV}$), which was $1.00 \pm 0.06$ (n = 4 patches) for activation by PI(3,5)P$_2$, $0.93 \pm 0.03$ for LyNA1 (n = 4), but only $0.10 \pm 0.03$ for LyNa-VA1.2 (n = 6, *Figure 2F*).

Under the voltage-step protocols, LyNa-VA-activated $I_{TPC2}$ exhibited substantial time-dependent inactivation at negative voltages (*Figure 2C and G* and *Figure 2—figure supplement 2A*). This could be conveniently characterized by two parameters: the amount of current decline at steady state ($\frac{I@500ms}{I@10ms}$) and the time constant ($\tau$) of the decay of current. There was no inactivation with ratio near 1.0 at all potentials when the agonist was PI(3,5)P$_2$ or LyNA1, but the ratio was 0.6 or less for all the active TCAs tested at $-120$ mV; the time constant of inactivation was approximately 300 ms at $-120$ mV (*Figure 2J* and *Figure 2—figure supplement 2*).

When we used a tail-current protocol to study the voltage-dependent activation, the activation by LyNa-VA1.2 was strongly dependent on voltage, with a half-maximal activation voltage (V$_{1/2}$) of $-20$ mV at the concentration of 100 µM (*Figure 2J and K*). In contrast, no apparent voltage-dependent activation was seen when whole-endolysosome $I_{TPC2}$ was activated by PI(3,5)P$_2$ (*Figure 2I and K*) or LyNA1. Collectively, LyNa-VAs have manifested multiple aspects of voltage-dependent gating of TPC2 channels.

## Synergistic activation of TPC2 by PI(3,5)P$_2$ and TCAs

PI(3,5)P$_2$ reportedly binds to Lys204 and adjacent AA residues to activate TPC2 (*She et al., 2019*). In the cells that were transfected with a PI(3,5)P$_2$-insensitive mutant TPC2 channel (K204A) (*She et al., 2019*), LyNa-VA and LyNA1 still robustly activated whole-endolysosome $I_{TPC2}$ (*Figure 3A and B* and *Figure 3—figure supplement 1A and B*). Lysosomal PI(3,5)P$_2$ may play a permissive role in TPC activation in intact cells (*Ruas et al., 2015*). Much dramatic LyNa-VA activation was seen in the presence of a low concentration of PI(3,5)P$_2$ (50 nM; *Figure 3C and D* and *Figure 3—figure supplement 1C*), and this synergism was nearly abolished in TPC2$^{K204A}$ mutant channels (*Figure 3C and D*). In contrast, an additive but not synergistic activation was observed between LyNA1 and PI(3,5)P$_2$ (*Figure 3—figure supplement 1D and E*).

Although LyNa-VAs weakly activated endogenous TPC currents, in the presence of PI(3,5)P$_2$, more robust activation was seen (*Figure 3E and G*). In contrast, no measurable whole-endosome $I_{TPC}$ was seen in TPC1/2 DKO HAP1 cells even in the presence of both PI(3,5)P$_2$ and LyNa-VAs (*Figure 3F and G*). Collectively, these results suggested that LyNa-VAs activated or modulated TPC2 via a unique, PI(3,5)P$_2$-independent but voltage-dependent mechanism.

## Voltage-dependent activation of TPC1 by TCAs

In contrast to TPC2, which produces little or no current at any potential under basal conditions, yet a large, voltage-independent conductance increases in the presence of PI(3,5)P$_2$, TPC1 shows substantial voltage-dependent currents in the absence of any exogenous agonist (*Cang et al., 2014*; *Guo et al., 2016*). It was therefore of interest to explore whether TCAs/LyNa-VAs could also act on TPC1. To investigate this, we used two different voltage paradigms. First, when membrane voltage was stepped to various test potentials after a prolonged pre-pulse to +80 mV and the currents were compared to the basal currents, LyNa-VA1.1 produced a profound suppression of outward currents at potentials positive to E$_{rev}$ (~0 mV), and a substantial enhancement of inward currents negative to E$_{rev}$ (*Figure 4A and B*). Hence, upon LyNa-VA1 activation, the I-V of TPC1 completely reversed from strong outward rectification to strong inward rectification (*Figure 4B and C*). Intriguingly, unlike LyNa-VA1.1 and LyNa-VA1.2, which activated inward $I_{TPC2}$ with EC$_{50}$s within micromolar ranges, both LyNa-VA2.1 and LyNA1 inhibited $I_{TPC1}$ (*Figure 4D* and *Figure 4—figure supplement 1C*). Second, we used a tail current paradigm to measure the amplitude of the peak inward current at $-120$ mV following prolonged steps to various potentials (*Figure 4E*). The effect of LyNa-VA1.1 was to

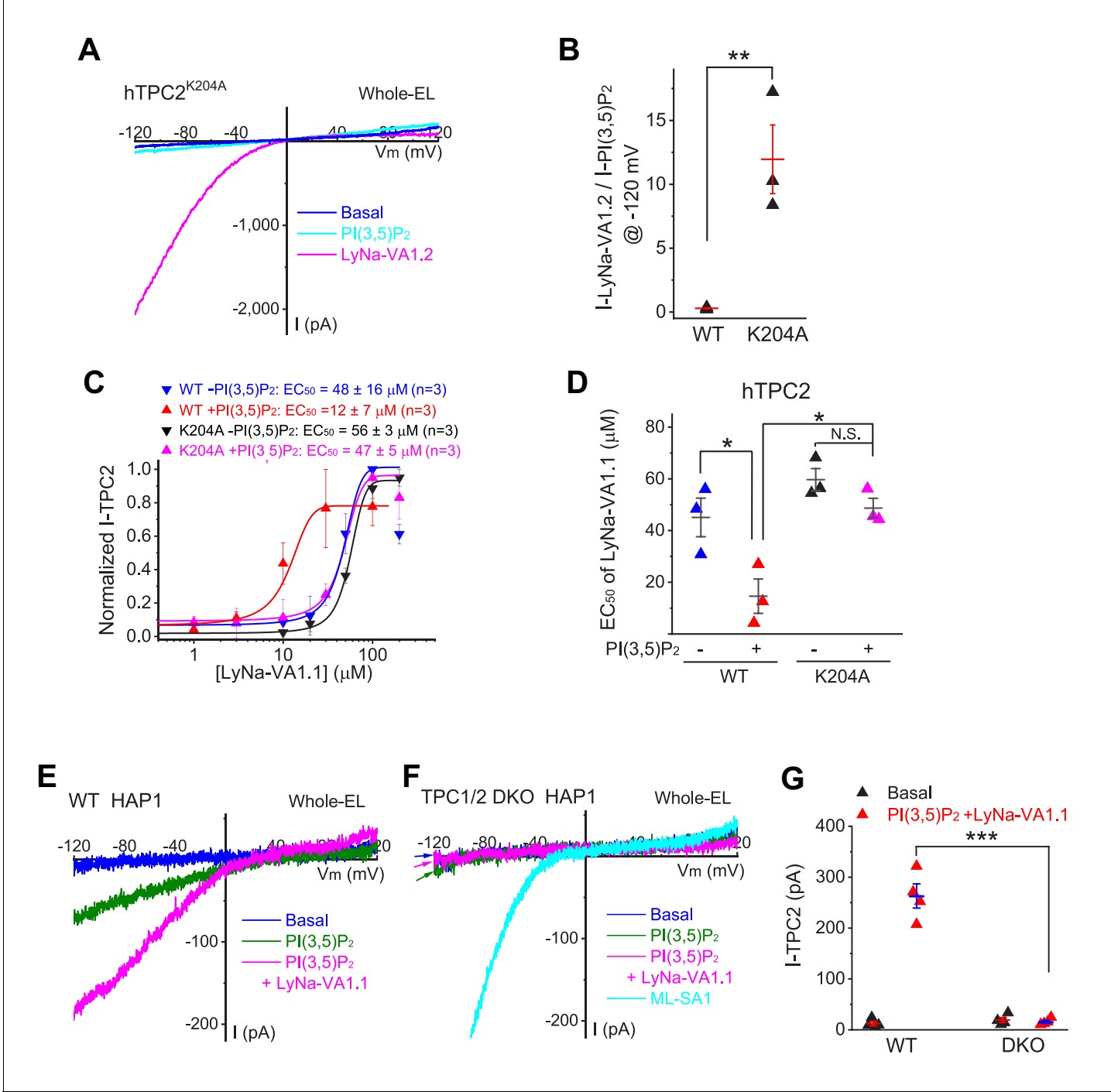

**Figure 3.** Synergistic activation of TPC2 channels by TCAs and PI(3,5)P$_2$. (**A**) The effects of PI(3,5)P$_2$ (0.3 µM) and LyNa-VA1.2 (100 µM) on whole-endolysosome $I_{TPC2-K204A}$ in TPC2$^{K204A}$-transfected HEK293 cells (*She et al., 2019*). (**B**) Comparison effects of LyNa-VA1.2 and PI(3,5)P$_2$ on WT and PI(3,5)P$_2$-insensitive K204A (*She et al., 2019*) mutant TPC2 channels (also see *Figure 3—figure supplement 1*). (**C**) The synergistic effects of PI(3,5)P$_2$ (50 nM) and LyNa-VA1.1 on whole-endolysosome $I_{TPC2}$ and $I_{TPC2-K204A}$ currents. Data are presented as Mean ± S.E.M (n = 3 patches). (**D**) The summary of EC$_{50}$ of LyNa-VA1.1 with or without PI(3,5)P$_2$ for WT and K204A mutant TPC2 channels. (**E, F**) Co-application of PI(3,5)P$_2$ (0.3 µM) and LyNa-VA1.1 (50 µM) activated whole-endolysosome $I$TPC in WT (**E**) but not TPC1/2 DKO (**F**) HAP1 cells. Note that the endogenous TPCs were more difficult to activate compared to overexpressed TPCs. (**G**) Summary of LyNa-VA1.1 effects on whole-endolysosome $I_{TPC}$ in WT and TPC1/2 DKO cells. For panels B, D and F, individual data and Mean ± S.E.M are presented. *, p<0.05; **, p<0.01; ***, p<0.001; N.S., no significance. Individual data for (**B–D**) and (**G**) are presented in *Figure 3—source data 1*.

The online version of this article includes the following source data and figure supplement(s) for figure 3:

*Figure 3 continued on next page*

*Figure 3 continued*

**Source data 1.** Synergistic activation of TPC2 channels by TCAs and PI(3,5)P$_2$.
**Figure supplement 1.** LyNa-VAs and LyNA1 activate TPCs independent of PI(3,5)P$_2$.
**Figure supplement 1—source data 1.** Dose-dependent activation of TPC2 by LyNA1 in the presence or absence of PI(3,5)P$_2$.

shift the midpoint of activation voltage (V$_{1/2}$) by −65 mV as compared to the basal condition, and to

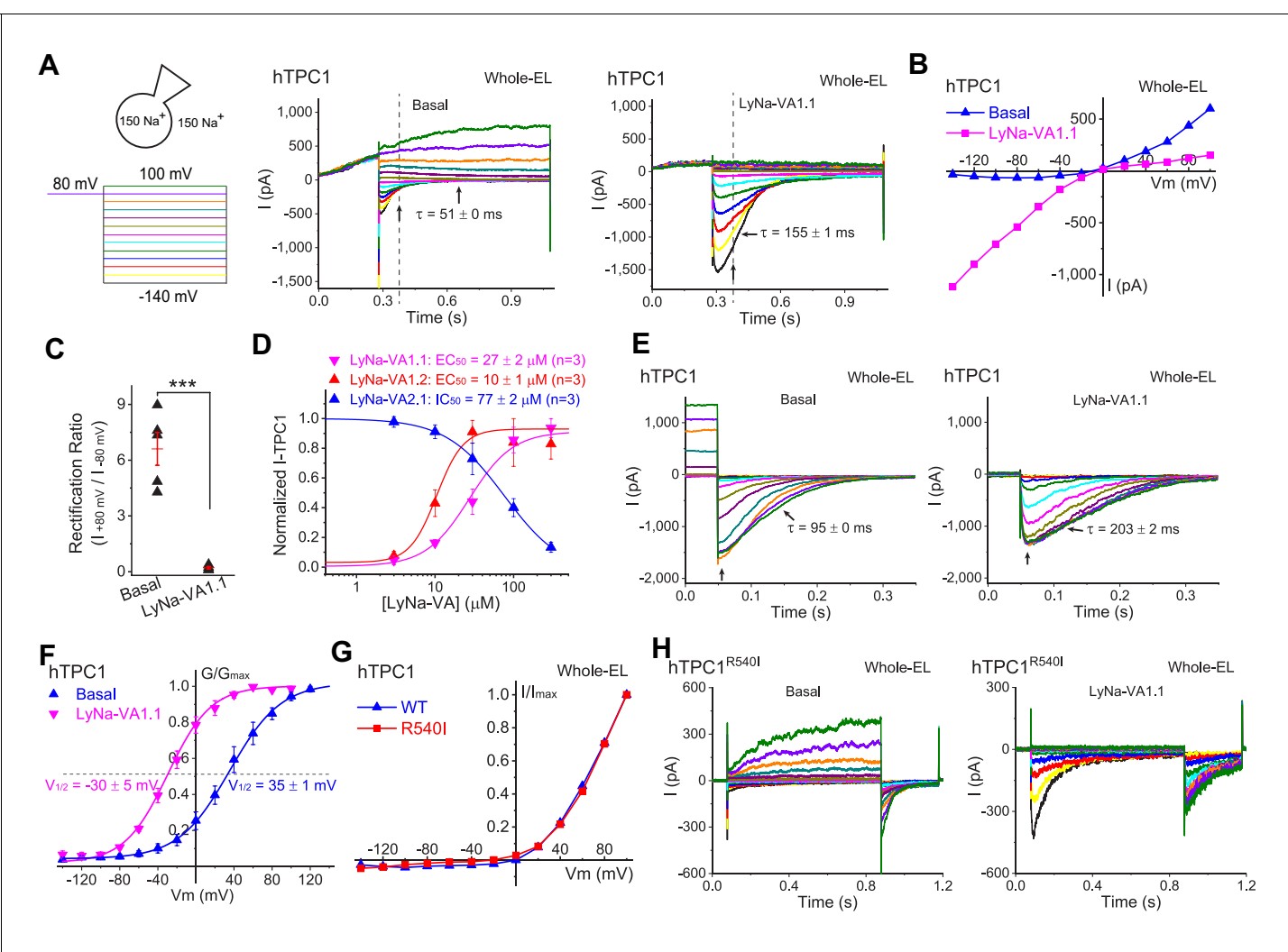

**Figure 4.** Activation of lysosomal TPC1 channels by LyNa-VAs. (**A**) Whole-endolysosome TPC1 current ($I_{TPC1}$) was activated by LyNa-VA1.1 (right) and elicited by a voltage step protocol (left), in which a preconditioning voltage (80 mV, 0.3 s) was applied before voltage steps starting from −140 to 100 mV (0.8 s, ΔV = 20 mV). HP = 0 mV. Unless otherwise indicated, symmetric 150 mM Na$^+$ solutions were used for all whole-endolysosome recordings, and LyNa-VA1.1 (100 μM) was bath-applied. (**B**) I-V plots of basal- and LyNa-VA1.1-induced $I_{TPC1}$, which were recorded from the same vacuole. Dotted lines in A indicate where the currents were measured. (**C**) Summary of rectification index of basal and LyNa-VA1.1-induced $I_{TPC1}$. Individual data and Mean ± S.E.M are presented. ***, p<0.001. (**D**) Does-dependent activation or inhibition of TPC1 by LyNa-VA1.1, LyNa-VA1.2, and LyNa-VA2.1. Data are presented as Mean ± S.E.M (n = 3 patches). (**E**) The effects of LyNa-VA1.1 on $I_{TPC1}$ tail currents at −120 mV. The voltage protocol that elicited $I_{TPC1}$ tail currents was shown in *Figure 4—figure supplement 1A*. (**F**) The effects of LyNa-VA1.1 on the G-V curves of $I_{TPC1}$ (n = 4–5 patches). (**G**) Normalized I-V plots of basal voltage-dependent currents in WT TPC1- and TPC1$^{R540I}$-transfected cells. (**H**) The basal and LyNa-VA1.1-activated $I_{TPC1-R540I}$ step currents, which were recorded from the same vacuole. Individual data for (**C**), (**D**) and (**F**) are also presented in *Figure 4—source data 1*.

The online version of this article includes the following source data and figure supplement(s) for figure 4:

**Source data 1.** Activation of lysosomal TPC1 channels by LyNa-VAs.
**Figure supplement 1.** Inhibition of lysosomal TPC1 channels by LyNA-1.
**Figure supplement 2.** Voltage-dependence and PI(3,5)P$_2$-sensitivity of R540I mutant TPC1 channels.

slow down the deactivation time course (*Figure 4E–F*).

The S4 segments of TPC1 and TPC2 contain several positively-charged AA residues, which were believed to serve as voltage sensors in mediating TPC1- but not TPC2-specific activation (*She et al., 2018*). It was recently reported that the R540I mutation, which removes a positive charge in the second putative voltage-sensing 'S4-type' helix, abolished TPC1 activation by membrane depolarization (*She et al., 2018*). However, in our hands, depolarization still robustly activated voltage-dependent currents in the whole-endolysosomes of hTPC1$^{R540I}$ -expressing cells (*Figure 4G and H*). When LyNa-VA1.1 was tested on endolysosomes overexpressing TPC1$^{R540I}$, there was also a dramatic enhancement of inward currents at negative potentials indicative of a substantial negative shift in the V$_{1/2}$ of voltage activation (*Figure 4H*). Finally, when exposed to PI(3,5)P$_2$, I$_{TPC1}$ showed a large enhancement at negative potentials indicative of a negative shift in the V$_{1/2}$ of activation, while TPC1$^{R540I}$ -positive endolysosomes showed large currents at all potentials (*Figure 4—figure supplement 2A, B and C*). The charge-introducing mutation at Ile551 of TPC2 (*She et al., 2019*), the equivalent site of TPC1$^{R540}$, conferred a voltage-dependence to the mutant channel but failed to affect LyNa-VA1.2 activation (*Figure 4—figure supplement 2D and E*). Put together, these results suggested that whereas the S4 voltage-sensing domains may play a modulatory role, there exist intrinsic or extrinsic voltage-sensing mechanisms elsewhere responsible for the voltage-dependent gating of TPCs.

## Cationic ion selectivity of TPC2 is not altered by LyNa-VAs and LyNA1

The selectivity filter region in an ion channel is responsible for the selective permeability to one or more ions (*Yu and Catterall, 2004*). However, it was recently reported that AA residues below the selectivity filter may mediate activation gating of multiple K$^+$ channels by small-molecule agonists (*Schewe et al., 2019*). In addition, some voltage sensitivity can be conferred when the permeable ions move through the selectivity filter (*Schewe et al., 2016*). We therefore tested whether TCAs/LyNa-VAs bind to the selectivity filer. Like PI(3,5)P$_2$-activated I$_{TPC2}$ (*Figure 5A–C and K*), LyNa-VA1.2- and LyNA1-induced whole-endolysosome I$_{TPC2}$ was highly selective for Na$^+$ over K$^+$, as substitution of bath/cytoplasmic Na$^+$ with K$^+$ significantly shifted E$_{rev}$ to more positive values (*Figures 1H*, *5D–F and K* and *Figure 2—figure supplement 1B*). Meanwhile, LyNa-VA1.2- and LyNA1-activated I$_{TPC2}$ still exhibited low Ca$^{2+}$ permeability (*Figure 5G and K* and *Figure 5—figure supplement 1A*), and the P$_{Ca}$/P$_{Na}$ values were similar to those for PI(3,5)P$_2$-activated I$_{TPC2}$ (*Wang et al., 2012*) (*Figure 5K* and *Figure 5—figure supplement 1B*). Finally, the N653G mutation, which is known to significantly increase the relative K$^+$ permeability of the TPC2 channel (*Guo et al., 2017*), did not alter the ability of LyNa-VA1.2 to enhance channel activation, but did result in currents elicited by LyNa-VA1.2 that had a much increased relative permeability to K$^+$ (*Figure 5I–K*). Taken together, these results indicate that the selectivity filter region does not mediate the action of TCAs.

## Discussion

In this study, we report that the voltage dependence, generally thought to be an intrinsic property of an ion channel (*Catterall, 2010*; *Yu and Catterall, 2004*), can be conferred or unmasked by extrinsic agonists in lysosomal TPC channels. What is the origin of agonist-conferred voltage-dependence in the otherwise voltage-independent TPC2 channel? It is possible that the S4 voltage sensors, which are operational in TPC1, might be 'exposed' upon agonist binding. Alternatively, another unidentified 'hidden' intrinsic voltage sensor might be revealed upon agonist binding, a possibility that was supported by the negative results in our targeted mutational analyses of the S4 regions. Additionally, it is recently reported that permeant ions may contribute to the channel's voltage-dependence (*Schewe et al., 2019*). Hence, it is an attractive hypothesis that a gate at the selectivity filter is the molecular target of TCAs to mediate the observed voltage-dependence. However, it remains unknown whether such a 'selectivity filter gate', extensively studied in several other ion channels including CNG channels (*Contreras et al., 2008*), does exist in TPC channels. Nevertheless, although mutational analyses in the selectivity filter or the 'S6 gate' (data not shown) do not seem to affect LyNa-VA activation, given the limitations of targeted mutations, future high-resolution structural studies may be necessary to reveal the TCA-binding sites in the TPC channels, explaining the conferred voltage dependence by TCAs. In addition, the differential effects of LyNa-VAs on TPC1

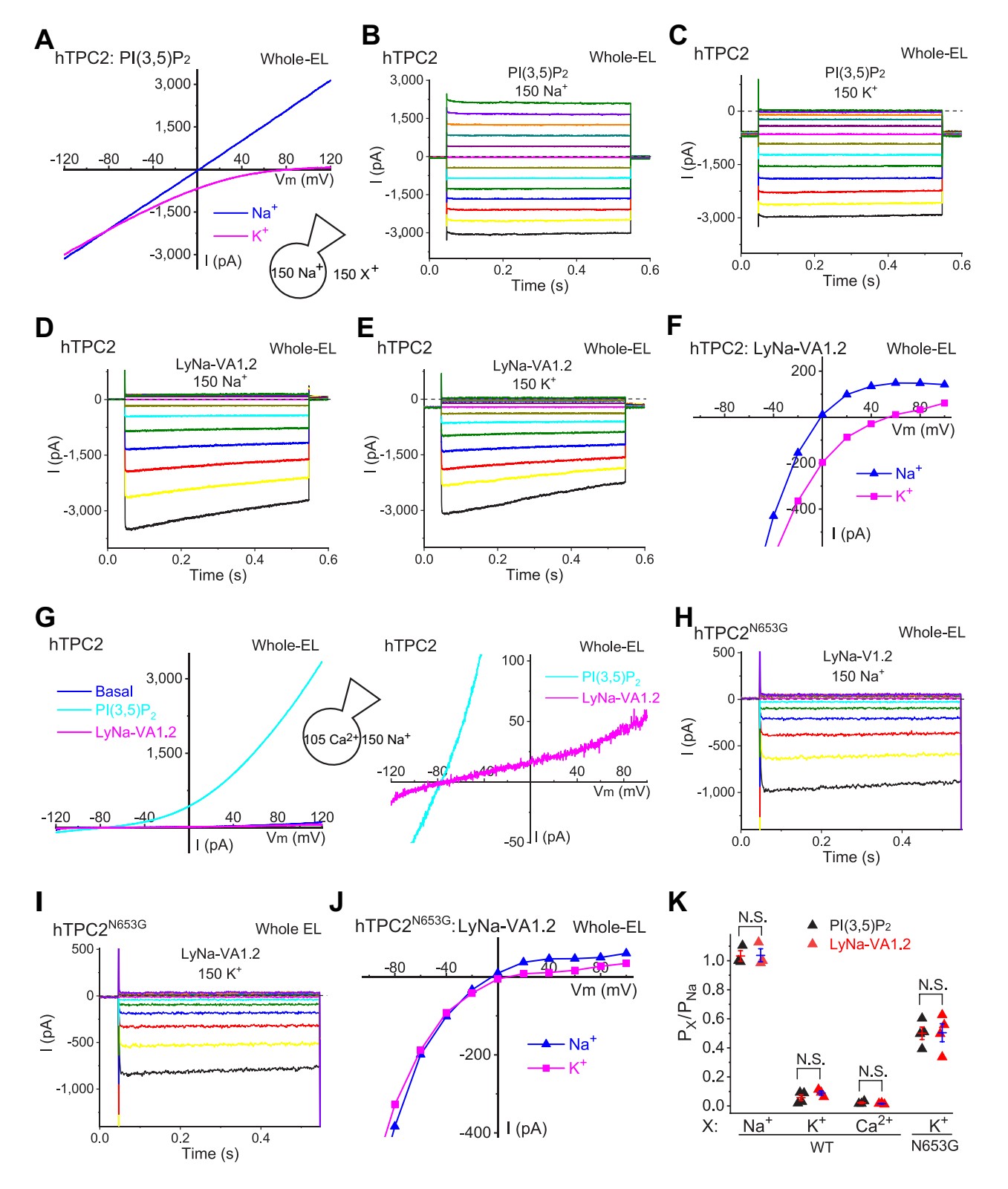

**Figure 5.** LyNa-VAs do not change the cationic selectivity of TPC2 channels. (**A**) Representative PI(3,5)P$_2$-evoked whole-endolysosome $I_{TPC2}$ elicited by a voltage ramp from −120 to 120 mV. The recordings were performed under a bi-ionic condition with 150 (in mM) Na$^+$ in the pipette solution and 150 Na$^+$ or 150 K$^+$ in the bath solution. PI(3,5)P$_2$ (0.3 µM) was bath-applied. (**B, C**) Representative PI(3,5)P$_2$- evoked $I_{TPC2}$ elicited with voltage steps (−140 mV to 100 mV with a ΔV = 20 mV, 0.5 s) with 150 mM Na$^+$ (**B**) or K$^+$(**C**) in the bath solution. (**D, E**) Representative LyNa-VA1.2-activated $I_{TPC2}$ step

*Figure 5 continued on next page*

*Figure 5 continued*

currents. (F) Representative I-V plots of LyNa-VA1.2- activated $I_{TPC2}$ measured from the instantaneous currents in (D) and (E). Note the reversal potentials ($E_{rev}$) of LyNa-VA1.2- activated $I_{TPC2}$ in the presence of Na$^+$ or K$^+$ bath solution. (G) LyNa-VA1.2- activated $I_{TPC2}$ under the bi-ionic conditions of bath/cytosolic Na$^+$ and pipette/luminal Ca$^{2+}$. 150 Na$^+$ solution contained (in mM) 145 NaCl, 5 NaOH, 20 HEPES, (pH 7.2); isotonic (105 mM) Ca$^{2+}$ solution contained (in mM) 100 CaCl$_2$, 5 Ca(OH)$_2$, 20 HEPES (pH 7.2). Right panel zoom-in micrograph shows the $E_{rev}$ of LyNa-VA1.2- activated $I_{TPC2}$. (H–J) Representative I-V plots of LyNa-VA1.2- evoked $I_{TPC2-N653G}$ (J) measured from Na$^+$ (H) and K$^+$ (I) bath solution. (K) Summary of Na$^+$ *vs.* K$^+$ /Ca$^{2+}$ selectivity of WT TPC2 and TPC2$^{N653G}$ channels. Individual data and Mean ± S.E.M are presented (also see *Figure 5—source data 1*). N.S., no significance.

The online version of this article includes the following source data and figure supplement(s) for figure 5:

**Source data 1.** Ionic selectivity of WT TPC2 and N653G mutant channels.
**Figure supplement 1.** LyNA1 does not change the Na$^+$ selectivity of TPC2 channels.
**Figure supplement 1—source data 1.** LyNA1 does not change ionic selectivity of TPC2 channels.

and TPC2 may also help design future structural-functional studies to reveal the action site of TCAs on TPCs.

Diverse functions have been associated with TPC channels, largely based on genetic manipulations (e.g. KO, knockdown, overexpression) (*Grimm et al., 2017*; *Xu and Ren, 2015*). However, the roles of TPCs in some of the proposed functions might be indirect based on the following reasons. First, lysosomal membrane trafficking, for example fusion and fission, has been difficult to study as these functions are interconnected. For instance, a block in membrane fusion may often indirectly affect membrane fission, and vice versa (*Xu and Ren, 2015*). In addition, defects in lysosomal membrane trafficking may also affect lysosomal degradation, and degradation products may in turn regulate membrane trafficking (*Xu and Ren, 2015*). Second, compensatory changes occur commonly in the genetically-manipulated cells, for example KO or lysosome storage disease (LSD) cells. Hence, it is necessary to develop methods to acutely activate and inhibit lysosomal channels, so that immediate cellular actions of TPC activation can be revealed. Notably, precisely defining TPC's roles in lysosomal $\Delta\psi$ regulation may require real-time monitoring of lysosomal $\Delta\psi$ while acutely activating or inhibiting lysosomal K$^+$ and Na$^+$ channels (*Li et al., 2019*). The identification of membrane-permeable small-molecule TPC agonists has made it feasible for such studies.

TCAs are known to regulate autophagy and lysosome function, but underlying mechanisms are not clear (*Tsvetkov et al., 2010*). For example, in a neuronal model of Huntington disease (HD), TCAs were shown to be neuroprotective by inducing the clearance of misfolded protein aggregates (*Tsvetkov et al., 2010*). Given the proposed roles of TPCs in autophagy and lysosomal membrane trafficking (*Grimm et al., 2017*; *Li et al., 2019*), it is likely some of the effects associated with TCAs are mediated by TPCs. The concentrations of TCAs that activate autophagy are lower than those activating TPC channels in the current study (*Beckmann et al., 2014*; *Cheng and Bahar, 2019*). However, TCAs, as lysosomotropic compounds, are known to accumulate at high concentrations in the lysosomes (*Beckmann et al., 2014*). In addition, the synergistic effects of TCAs with endogenous ligand, for example PI(3,5)P$_2$, suggest that the exposure level of TCAs in the brain might be sufficient to cause robust pharmacological actions, much more potently than the efficacies defined in our channel assays. Future cell biological studies utilizing TCAs with TPC KO as negative controls may confirm whether TCAs modulate autophagy and lysosome function through activation of TPCs.

## Materials and methods

### Key resources table

| Reagent type (species) or resource | Designation | Source or reference | Identifiers | Additional information |
|---|---|---|---|---|
| Cell line (*Homo sapiens*) | HEK293 | ATCC | RRID:CVCL_0045 | |
| Cell line (*Homo sapiens*) | HAP1 | Horizon Discovery | Cat. #: C631 | |

*Continued on next page*

*Continued*

| Reagent type (species) or resource | Designation | Source or reference | Identifiers | Additional information |
|---|---|---|---|---|
| Cell line (*Chlorocebus aethiops*) | Cos1 | ATCC | RRID:CVCL_0223 | |
| Recombinant DNA reagent | pEGFP-C2-TPC2 (plasmid) | *Wang et al., 2012* | | |
| Recombinant DNA reagent | pEGFP-C2-TPC1 (plasmid) | *Wang et al., 2012* | | |
| Sequence-based reagent | TPC1-sgRNA | This paper | sgRNA | CTTGCAGTACTTCAGCACCC |
| Sequence-based reagent | TPC2-sgRNA | This paper | sgRNA | CCCCAGCGTCGGGCTGCTGC |
| Sequence-based reagent | TPC1-fw | This paper | PCR primers | ATGGCCCAGACATGTGACTC |
| Sequence-based reagent | TPC1-re | This paper | PCR primers | TGCCTGTCTCCATCCTCTCA |
| Sequence-based reagent | TPC2-fw | This paper | PCR primers | TGAGCTGAGCATGAGGCAAG |
| Sequence-based reagent | TPC2-re | This paper | PCR primers | AAAGGACAAGTGGCCCTGAG |
| Chemical compound, drug | Desipramin hydrochloride | Sigma | Cat. #: D3900 | |
| Chemical compound, drug | Carbamazepine | Sigma | Cat. #: C4024 | |
| Chemical compound, drug | monensin | Sigma | Cat. #: M5273 | |
| Chemical compound, drug | ionomycin | Sigma | Cat. #: I0634 | |
| Chemical compound, drug | Clomipramine | Cayman Chemical | Cat. #: 15884 | |
| Chemical compound, drug | Imipramine | Cayman Chemical | Cat. #: 15890 | |
| Chemical compound, drug | Amitriptyline | Cayman Chemical | Cat. #: 15881 | |
| Chemical compound, drug | Nortriptyline | Cayman Chemical | Cat. #: 15904 | |
| Chemical compound, drug | Phenothiazine | NCATS | CAS: 92-84-2 | |
| Chemical compound, drug | Triflupromazine | NCATS | CAS: 146-54-3 | |
| Chemical compound, drug | Chlorpromazine | Cayman Chemical | Cat. #: 16129 | |
| Chemical compound, drug | ML-SA1 | Princeton BioMolecular Research Inc | Cat. #: OSSK_389119 | |
| Chemical compound, drug | $PI(3,5)P_2$ | Echelon Biosciences | Cat. #: P-3508 | |
| Chemical compound, drug | vacuolin-1 | Calbiochem | Cat. #: 673000 | |
| Software, algorithm | pClamp | pClamp | RRID:SCR_011323 | |

## Molecular biology

All TPC1 and TPC2 mutants were generated with a site-directed mutagenesis kit (Qiagen) using EGFP- human TPC1 and TPC2 as the templates. All constructs were confirmed by DNA sequencing.

## Mammalian cell lines

Cos1 (ATCC, CRL-1650, passage number 8–15) and HEK293 (ATCC, CRL-1573, passage number 8–15) cells were cultured in DMEM with 10% fetal bovine serum (FBS). HAP1 (Horizon Discovery, human HAP1 parental cell line, passage number 5–15) cells were maintained in IMDM with 10% FBS. Immortalized cell lines (Cos1, HEK293 and HAP1) were cultured following standard tissue culture protocols, and were tested negative for mycoplasma contamination using MycoAlert Mycoplasma Detection Kit Assay (Lonza). HEK293 cells are on the list of frequently misidentified or cross-contaminated cell lines, but were only used for the overexpression experiments. Cells were transfected with Lipofectamine 2000 (Invitrogen). After culture media were refreshed 18–24 hr post-transfection, cells were used for the electrophysiological assays 24–36 hr post-transfection.

## TPC1/TPC2 DKO HAP1 cells

TPC1 and TPC2 CRISPR KO cells were generated in HAP1 cells using the CRISPR/Cas9 system. The TPC1 sequence (5'cttgcagtacttcagcaccc3', TPC1-sgRNA) and TPC2 sequence (5'ccccagcgtcgggctgctgc3', TPC2-sgRNA) were targeted with pSpCas9 (BB)−2A-puro vector (Addgene). HAP1 cells were co-transfected with TPC1 and TPC2-sgRNA expressing vector in the presence of Lipofectamine 2000 (In vitrogen) and selected with 2 µg/ml puromycin for 48 hr. The remaining cells were trypsinized and seeded into 96-well plate after a limiting dilution. When single cell clones were established, their genomic DNAs were extracted and amplified with following primers: TPC1-fw: 5'atggcccagacatgtgactc3', TPC1-re: 5' tgcctgtctccatcctctca3'; TPC2-fw: 5'tgagctgagcatgaggcaag3', TPC2-re: 5' aaaggacaagtggccctgag3'. The PCR amplicons were then sequenced to confirm the intended genetic disruption. The TPC1/TPC2 DKO cells harboring an insert of one nucleotide (A) in the TPC1 sequence and a deletion of ten nucleotides in the TPC2 sequence, respectively, were used in the present study.

## LOPAC high-throughput screening

$Ca^{2+}$ imaging-based HTS using the Library of Pharmacologically Active Compounds (LOPAC)1280 collection (Sigma) was conducted as described previously (*Liu et al., 2010*). Briefly, HEK293 cells stably expressing TPC2 and TRPML1 were loaded with a $Ca^{2+}$ detection dye (Fluo-4). The kinetic of $Ca^{2+}$ flux was measured using a kinetic plate reader FDSS-7000 (*Liu et al., 2010*). The FDSS-7000 had an on-board 1536 pintool that was used to transfer 23 nl of a compound to the assay plate (*Liu et al., 2010*). All of the compounds were dissolved in 100% DMSO. Hence, transferring 23 nl of a compound to a well containing 5 µl of the culture medium would result in a final concentration of DMSO of ~0.5%. The final concentrations of the LOPAC compounds in each were 0.003, 0.015, 0.074, 0.37, 1.84, 9.2, and 46 µM, respectively. Compounds that were positive in the TPC2 assay, but not in the TRPML1 assay, were considered to be positive hits for potential TPC2 agonists.

## Whole-cell electrophysiology

Whole-cell recordings were performed using pipette electrodes with resistance of 3–5 MΩ. Unless otherwise stated, both pipette and bath solutions contained (in mM): 145 NaOH, 5 NaCl, 20 HEPES, pH 7.2 (adjusted with methanesulfonic acid). All bath solutions were applied via a perfusion system that allowed complete solution exchange within a few seconds. Data were collected using an Axopatch 200A patch clamp amplifier, Digidata 1440, and pClamp 10.2 software (Axon Instruments). Whole-cell currents were digitized at 10 kHz and filtered at 2 kHz. All experiments were conducted at room temperature (21–23°C), and all recordings were analyzed with pClamp 10.2, and Origin 8.0 (OriginLab, Northampton, MA).

## Whole-endolysosome electrophysiology

Endolysosomal electrophysiology was performed in isolated enlarged endolysosomes using a modified patch-clamp method (*Dong et al., 2010*). Cells were treated with 1 µM vacuolin-1, a lipid-soluble polycyclic triazine that can selectively increase the size of endosomes and lysosomes (*Huynh and Andrews, 2005*), for at least 1 hr and up to 12 hr. Whole-endolysosome recordings were performed on manually isolated enlarged endolysosomes (*Wang et al., 2012*). In brief, a patch pipette was pressed against a cell and quickly pulled away to slice the cell membrane. Enlarged endolysosomes were released into a dish and identified by monitoring EGFP-TPC1/2 or the mCherry-TPC1/2

fluorescence. After formation of a gigaseal between the patch pipette and the enlarged endolysosome, capacitance transients were compensated. Voltage steps of several hundred mVs with millisecond duration were then applied to break into the vacuolar membrane. The whole-endolysosome configuration was verified by the re-appearance of capacitance transients after break-in.

Unless otherwise stated, both bath (internal/cytoplasmic) and pipette solutions contained (in mM): 145 NaOH, 5 NaCl, 20 HEPES, pH 7.2 (adjusted with methanesulfonic acid). 150 $K^+$ solution contained (in mM): 145 KOH, 5 KCl, 20 HEPES, pH 7.2 (adjusted with methanesulfonic acid). Data were collected using an Axopatch 200A patch clamp amplifier, Digidata 1440, and pClamp 10.2 software (Axon Instruments). Whole-endolysosome currents were digitized at 10 kHz and filtered at 2 kHz. All experiments were conducted at room temperature (21–23°C), and all recordings were analyzed with pClamp 10.2, and Origin 8.0 (OriginLab, Northampton, MA). The permeability to cations (relative to $P_{Na}$) was estimated based on following equations (*Lewis, 1979*) and $E_{rev}$ measurement under bi-ionic conditions:

$$P_X/P_{Na} = \gamma_{Na}/\gamma_X \left\{ [Na^+]_{Luminal} / [X^+]_{Cytoplasmic} \right\} \{\exp(R_{rev}F/RT)\} \tag{1}$$

$$P_{Ca}/P_{Na} = \gamma_{Na}/\gamma_{Ca} \left\{ [Na^+]_{Cytoplasmic} / [4[Ca^{2+}]_{Luminal}] \right\} \{\exp(E_{rev}F/RT)\} \{1 + \exp(E_{rev}F/RT)\} \tag{2}$$

where R, T, F, $E_{rev}$, and $\gamma$ are, respectively, the gas constant, absolute temperature, Faraday constant, reversal potential, and activity coefficient. The liquid junction potentials were measured and corrected as described (*Neher, 1992*).

## Data analysis

Data are presented as the mean ± standard error of the mean (S.E.M). Statistical comparisons were made using analysis of variance (ANOVA). A p value < 0.05 was considered statistically significant.

## Acknowledgements

This work was supported by NIH RO1 grants (NS062792 and DK115474 to HX). Additional support was provided by an M-Cubed grant and a PFD initiative grant from the University of Michigan. A sponsored research grant from CalyGene Biotechnology provided interim funding for research supplies. The funders had no role in study design, data collection and analysis, decision to publish, or preparation of the manuscript. Research at NCATS was funded by the NIH intramural Research Program. We are grateful to Dr. Richard Hume for comments on an earlier version of the manuscript. We appreciate the encouragement and helpful comments from other members of the Xu laboratory.

## Additional information

### Funding

| Funder | Grant reference number | Author |
| --- | --- | --- |
| National Institute of Neurological Disorders and Stroke | NS062792 | Haoxing Xu |
| National Institute of Diabetes and Digestive and Kidney Diseases | DK115474 | Haoxing Xu |

The funders had no role in study design, data collection, and interpretation, or the decision to submit the work for publication.

### Author contributions

Xiaoli Zhang, Conceptualization, Data curation, Formal analysis, Validation, Investigation, Writing—original draft, Writing—review and editing; Wei Chen, Conceptualization, Data curation, Formal analysis, Validation, Investigation, Writing—review and editing; Ping Li, Conceptualization, Data curation, Investigation, Writing—review and editing; Raul Calvo, Marc Ferrer, Juan Jose Marugan,

Resources, Investigation, Writing—review and editing; Noel Southall, Xin Hu, Melanie Bryant-Genevier, Resources, Investigation; Xinghua Feng, Meimei Yang, Validation, Investigation; Qi Geng, Kaiyuan Tang, Validation, Investigation, Generated TPC1/2 mutant plasmids; Chenlang Gao, Validation, Investigation, Writing—review and editing, Generated TPC1/2 mutant plasmids; Haoxing Xu, Conceptualization, Resources, Supervision, Funding acquisition, Writing—original draft, Writing—review and editing

### Author ORCIDs
Xiaoli Zhang https://orcid.org/0000-0002-0507-6317
Wei Chen https://orcid.org/0000-0002-2794-2407
Noel Southall http://orcid.org/0000-0003-4500-880X
Juan Jose Marugan http://orcid.org/0000-0002-3951-7061
Haoxing Xu https://orcid.org/0000-0003-3561-4654

### Decision letter and Author response
Decision letter https://doi.org/10.7554/eLife.51423.sa1
Author response https://doi.org/10.7554/eLife.51423.sa2

## Additional files

### Supplementary files
• Transparent reporting form

### Data availability
All data generated or analyzed during this study are included in the manuscript and supporting files. Source data files have been provided for: Fig. 1A, 1E, 1G, 2F-G, 3B-D, 3G, 4C-D, 4F, 5K; Table 1; Fig. 1-figure supplement 2, Fig. 2- figure supplement 2C, Fig. 3 -figure supplement 1E and Fig. 5 - figure supplement 1B. LOPAC screening results in Fig. 1A has been deposited in Dyrad: DOI: https://doi.org/10.5061/dryad.s5f6j9h.

The following dataset was generated:

| Author(s) | Year | Dataset title | Dataset URL | Database and Identifier |
|---|---|---|---|---|
| Xu H, Zhang X, Chen W, Li P, Calvo R, Southall N, Hu X, Bryant-Genevier M, Feng X, Geng Q, Gao C, Yang M, Tang K, Ferrer M, Marugan J | 2019 | Data from: Agonist-specific Voltage-dependent Gating of Lysosomal Two-pore Na + Channels | https://doi.org/10.5061/dryad.s5f6j9h | Dyrad Digital Repository, 10.5061/dryad.s5f6j9h |

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
