## [Decision Letter]

Acceptance summary:

The function of lysosomes is more and more being linked to several important normal and pathological physiological states. Several of these functions are linked to ion transport mechanisms mediated by ion channels that are not completely understood. In this manuscript, the authors have identified small-molecule agonist for the lysosomal two-pore channels, TPC1 and TPC2. Strikingly these agonist are tricyclic compounds that have traditionally been used as antidepressants. This is a clear example of re-purposing of clinically-relevant compounds and it also open the window to understanding secondary effects of these drugs that cannot be explained by their classically described actions. Lysosomal ion channels are notoriously difficult to study, given their location and poorly characterized biophysical properties. The findings in this manuscript also suggest that these compounds could help identify drug-specific modes of activation in two-pore channels. Given the growing recognition of the importance of lysosomes in several diseases, this findings are of general interest to clinicians and basic scientist alike. The compounds identified and initially characterized here should make the task of understanding the physiology of TPC1 and TPC2 channels a little easier.

**Decision letter after peer review:**

Thank you for submitting your article "Agonist-specific voltage-dependent gating of lysosomal two-pore Na^+^ channels" for consideration by *eLife*. Your article has been reviewed by three peer reviewers, including Leon D Islas as the Reviewing Editor and Reviewer #1, and the evaluation has been overseen by Richard Aldrich as the Senior Editor. The following individuals involved in review of your submission have agreed to reveal their identity: Shmuel Muallem (Reviewer #2); Tinatin Brelidze (Reviewer #3).

The reviewers have discussed the reviews with one another and the Reviewing Editor has drafted this decision to help you prepare a revised submission.

Summary:

The paper by the Xu group describes the novel finding that tricyclic anti-depressants, a major class of clinically relevant drugs, also behave as activators of the two-pore, Na-selective lysosomal ion channels TPC1 and TPC2. The authors show that these drugs induce voltage-dependent and voltage-independent modes of activation of these channels. These findings are relevant because they provide a possible new framework for the development of new TPC-selective activators (and possibly, inhibitors) with increased potency. These drugs offer the possibility of having an increased toolset for the study of these class of channels and open the door for understanding side effects and off-target effects of tricyclic anti-depressants. The reviewers are very enthusiastic about the manuscript and offer the following comments that the authors should attend.

Major comments:

1) While the authors showed that the effect is specific to TPC channels the effect may not be direct and might be mediated by a signaling cascade. This could be determined by recording currents from excised patches in the absence and presence of the agonists. In addition, the comparison of the time-course of activation for inside-out and whole cell or outside-out patches may suggest if the agonist binding site is extracellular or intracellular.

2) For pharmacological comparisons it would be useful to determine the EC_50_ for TPC1 and at least a few of the identified agonists.

3) Figure 3C could be strengthened by including dose-response data for K204 mutant. It is conceivable that the agonists might activate or expose additional PIP2 sites, distinct from the one destabilized by the K204A mutation. The dose-response experiments might reveal this possibility.

4) The effect of Riluzole is also very interesting and could be considered in more depth. The authors have not described the effect of Riluzole on TPC1 channels. Is it still voltage-independent? It is important to know if the mechanism of this agonist action is similar in the two channels, as it might help in identifying the molecular determinants of the effect. It also would be of interest to investigate the effect of Riluzole on TPC2 and R540I TPC1 mutant channels in the presence of PIP2. Both agonists show linear activation with voltage and therefore there is a possibility that they might be competitive.

5) Since the hTPC2 currents activated by TCAs are so different (Figure 2C), the possibility exists that channels other than TPC2 mediate these currents. The authors should endeavor to show that an hTPC2-specific blocker also blocks TCA-activated currents.

6) In the subsection “Cationic ion selectivity of TPC2 is not altered by LyNa-Vas”, the authors make the argument that to probe if TCAs bind to the selectivity filter, they investigate possible changes in ion selectivity. They find that there are no changes in selectivity in the presence of TCAs and therefore these drugs do not bind at the selectivity filter. This argument is flawed; there is no reason to think that a drug binding in or near the selectivity filter necessarily alters the selectivity to ions.

7) Further, the authors mention a "selectivity gate" do they mean "gate at the selectivity filter"? If so, there is no evidence that TPC channel's gating occurs at the selectivity filter, like in CNG cation channels. It is sometimes wrongly assumed that all ions have gating at a bundle crossing (S6 gate) *and* at the selectivity filter.

8) The last paragraph of the Discussion presents some discussion of the possible consequences of this newly described interaction between TCAs and lysosomal cation channels, however, there is no discussion of the possible mechanism of voltage-dependent gating induced in TPC channels and this is a major point of the manuscript.

9) In Figure 1E, it is surprising that the magnitude of the currents is so similar between cells to have the size of errors presented. Surely there should be a large variability in the levels of expression. Please present all comparisons between currents as normalized current density (current/capacitance).

---

## [Author Response]

Although the reviewers found our work interesting, they have made several specific suggestions to improve the paper. To address the reviewers’ concerns, we have performed a number of new experiments. The most significant changes are highlighted in the summary paragraphs below, and all other concerns are addressed in the point-by-point response.

First, in response to the reviewers’ suggestions, we performed additional new experiments to systematically investigate the effects of LyNA1 (the voltage-independent synthetic agonist) on both TPC1 and TPC2 channels, especially regarding the ion selectivity and synergistic activation with PI(3,5)P_2_. Unlike LyNa-VAs (the voltage-dependent synthetic agonists), LyNA1 activated TPC2 currents (*I*_TPC2_) in a voltage-independent manner (Figure 2D, 2E) through an additive but not synergistic mechanism to PI(3,5)P_2_ (new Figure 3—figure supplement 1D, E). In contrast, *I*_TPC1_ was inhibited by LyNA1 (new Figure 4—figure supplement 1B). Neither LyNa-VAs nor LyNA1 affected the Na^+^-selectivity of TPC channels (Figure 5K and new Figure 5—figure supplement 1). Hence, LyNA1 and LyNa-VAs modulate TPCs via distinct mechanisms.

Second, based on the reviewers’ suggestions, we explored the possible action site of LyNa-VAs by comparing the time course of activation in three different patch configurations: whole-cell (i.e., extracellular agonist application), inside-out (intracellular agonist application), and whole-endolysosome (intracellular agonist application). As channel activation in the whole-cell configuration was significantly slower, i.e., longer latency and time constant of activation, in comparison to cytosolic applications under the inside-out and whole-endolysosome configurations (see new Figure 1—figure supplement 2), the action site of LyNa-VAs is likely to be either intracellular or more accessible from the intracellular side.

Third, the reviewers asked us to examine the effects of multiple LyNa-VAs on TPC1 and various mutant TPC2 channels, including I551R, the presumed voltage-sensor mutation (She et al., 2018), and K204A, the PI(3,5)P_2_-binding-site mutation (She et al., 2019). We found that whereas both LyNa-VA1.1 and LyNa-VA1.2 activated TPC1, LyNa-VA2.1 instead inhibited TPC1 (new Figure 4D). Additionally, the I551R mutation, which converted TPC2 into a voltage-dependent outward-rectifying TPC1-like channel (She et al., 2018), failed to affect the voltage-dependent activation of inwardly-rectifying currents by LyNa-VAs (new Figure 4—figure supplement 2D-E). Furthermore, the lack of a synergistic effect between LyNa-VAs and PI(3,5)P_2_ on TPC2^K204A^ currents (new Figure 3C, 3D) suggested that Lys204 is the primary PI(3,5)P_2_-binding site. Collectively, the results obtained from these additional experiments have significantly strengthened the major conclusion of the paper: LyNa-VAs bind to activate TPCs via a voltage-dependent mechanism independent of PI(3,5)P_2_.

Major comments:1) While the authors showed that the effect is specific to TPC channels the effect may not be direct and might be mediated by a signaling cascade. This could be determined by recording currents from excised patches in the absence and presence of the agonists. In addition, the comparison of the time-course of activation for inside-out and whole cell or outside-out patches may suggest if the agonist binding site is extracellular or intracellular.

Based on the reviewer’s suggestion, we performed inside-out patch-clamp recordings in TPC2^LL/AA^-expressing HEK293 cells. Robust activation was seen in these cell-free excised patches (Figure 1—figure supplement 2), as well as in (cell-free) whole-endolysosome patches, suggesting that the effects of LyNa-VAs are most likely to be direct. As mentioned in the summary paragraphs (Experiment #2), LyNa-VAs activated *I*_TPC2_ significantly faster (i.e., with shorter latency and smaller activation time-constant) in the inside-out and whole-endolysosome patches (Figure 1—figure supplement 2C and D) comparted with whole-cell recordings, suggesting that the agonist binding site is likely to be intracellular.

2) For pharmacological comparisons it would be useful to determine the EC_50_ for TPC1 and at least a few of the identified agonists.

We have conducted the dose-dependent studies to show that whereas LyNa-VA1.1 and LyNa-VA1.2 activated whole-endolysosome*I*_TPC1_ with an EC_50_ of 27 ± 2 µM (n=3 patches) and 10 ± 1 µM (n=3), respectively, LyNa-VA2.1 inhibited *I*_TPC1_ (IC_50_ = 77 ± 2 µM, n=3; see Figure 4D). Also see Experiment #3 in the summary paragraphs.

3) Figure 3C could be strengthened by including dose-response data for K204 mutant. It is conceivable that the agonists might activate or expose additional PIP2 sites, distinct from the one destabilized by the K204A mutation. The dose-response experiments might reveal this possibility.

We agree with the reviewer and have examined the dose-dependent responses of TPC2^K204A^ in the presence and absence of PI(3,5)P_2_. Unlike wild-type (WT) channels, PI(3,5)P_2_ had little or no effect on the LyNa-VA dose-response of the TPC2^K204A^ channel (new Figure 3C and D), suggesting that Lys204 is essential for PI(3,5)P_2_ binding in TPC2 (She et al., 2019) in the presence and absence of LyNa-VAs.

4) The effect of Riluzole is also very interesting and could be considered in more depth. The authors have not described the effect of Riluzole on TPC1 channels. Is it still voltage-independent? It is important to know if the mechanism of this agonist action is similar in the two channels, as it might help in identifying the molecular determinants of the effect. It also would be of interest to investigate the effect of Riluzole on TPC2 and R540I TPC1 mutant channels in the presence of PIP2. Both agonists show linear activation with voltage and therefore there is a possibility that they might be competitive.

Whereas TPC2 was activated by Riluzole (LyNA1) with an EC_50_ of 181 ± 6 µM (Figure 3—figure supplement 1E), TPC1 was, intriguingly, inhibited by Riluzole at the concentration of 150 µM (Figure 4—figure supplement 1C). Since the inhibition appeared to be voltage-independent (Figure 4—figure supplement 1C), we did not further study the Riluzole effect on TPC1^R540I^. On the other hand, although both PI(3,5)P_2_ and Riluzole are agonists of TPC2, PI(3,5)P_2_ did not further increase the potency of Riluzole (EC_50_ = 175 ± 19 µM) on *I*_TPC2_ (Figure 3—figure supplement 1E). Additionally, Riluzole activation was intact in TPC2^K204A^ channels (Figure 3—figure supplement 1B). Therefore, Riluzole and PI(3,5)P_2_ may activate TPC2 channels via distinct voltage-independent mechanisms.

5) Since the hTPC2 currents activated by TCAs are so different (Figure 2C), the possibility exists that channels other than TPC2 mediate these currents. The authors should endeavor to show that an hTPC2-specific blocker also blocks TCA-activated currents.

We addressed the reviewer’s concern on this by conducting several control experiments. First, we used a pharmacological approach to show that TCA-activated currents were inhibited by our unpublished, TPC-specific small-molecule inhibitor (LyNI-1; Lysosomal Na^+^ channel Inhibitor 1; see Author response image 1). Note that Ned-19 and Tetrandine, two reported TPC blockers (Sakurai et al., 2015), are at best very weak inhibitors of *I*_TPC_ in our hands (Author response image 1). Likewise, verapamil, a blocker that we reported previously, had very weak inhibitory effects on the inward currents of TPC2 (Wang et al., 2012). We will report the characterization work of LyNI-1 in a separate study focusing on the cell biological roles of TPC2. Second, we used a genetic approach to show that TCA-activated whole-endolysosome currents were present in WT, but not TPC1/2 DKO HAP1 cells (Figure 3E and F). Third, TCA-activated whole-cell currents were only detected in TPC2^LL/AA^-transfected, but not non-transfected HEK293 cells (Figure 1C and D). Fourth, TCA-activated currents had nearly identical P_K_/P_Na_ values compared with PI(3,5)P_2_-activated TPC2 currents (Figure 5), and the N653G mutation decreased the Na^+^-selectivity of both TCA- and PI(3,5)P_2_-activated TPC currents (Figure 5 andFigure 5—figure supplement 1). Taken together, these results suggest that TCA-induced currents are mediated by TPC2.

**Author response image 1. respfig1:** LyNa-VA-induced currents are inhibited by LyNI-1, a small-molecule TPC2 inhibitor. (**A**) Dose-dependent inhibition of LyNI-1 on PI(3,5)P_2_ (0.1 µM)-induced whole-endolysosome *I*_TPC2_ (IC_50_ = 3 ± 1 µM, n=3 patches). (**B**) Co-application of LyNI-1(100 µM) inhibited LyNa-VA1.1 (50 µM)-evoked whole-endolysosome *I*_TPC2_.

6) In the subsection “Cationic ion selectivity of TPC2 is not altered by LyNa-Vas”, the authors make the argument that to probe if TCAs bind to the selectivity filter, they investigate possible changes in ion selectivity. They find that there are no changes in selectivity in the presence of TCAs and therefore these drugs do not bind at the selectivity filter. This argument is flawed; there is no reason to think that a drug binding in or near the selectivity filter necessarily alters the selectivity to ions.

We agree with the reviewer and have toned down the argument in the revision (see Discussion).

7) Further, the authors mention a "selectivity gate" do they mean "gate at the selectivity filter"? If so, there is no evidence that TPC channel's gating occurs at the selectivity filter, like in CNG cation channels. It is sometimes wrongly assumed that all ions have gating at a bundle crossing (S6 gate) and at the selectivity filter.

We agree with the reviewer and have modified the discussion about the unverified “gate at the selectivity filter” of TPC channels (see Discussion).

8) The last paragraph of the Discussion presents some discussion of the possible consequences of this newly described interaction between TCAs and lysosomal cation channels, however, there is no discussion of the possible mechanism of voltage-dependent gating induced in TPC channels and this is a major point of the manuscript.

We have expanded the discussion on the potential mechanisms that may contribute to TCA-induced voltage-dependent gating (see the first paragraph in Discussion).

9) In Figure 1E, it is surprising that the magnitude of the currents is so similar between cells to have the size of errors presented. Surely there should be a large variability in the levels of expression. Please present all comparisons between currents as normalized current density (current/capacitance).

Based on the reviewer’s suggestion, we have replaced the figure panel using normalized current density (new Figure 1E).